# A Deletion Encompassing the Furin Cleavage Site in the Spike Encoding Gene Does Not Alter SARS-CoV-2 Replication in Lung Tissues of Mink and Neutralization by Convalescent Human Serum Samples

**DOI:** 10.3390/pathogens11101152

**Published:** 2022-10-06

**Authors:** Fabrizia Valleriani, Lucija Jurisic, Chiara Di Pancrazio, Roberta Irelli, Eugenia Ciarrocchi, Michele Martino, Antonio Cocco, Elisabetta Di Felice, Maria Loredana Colaianni, Nicola Decaro, Barbara Bonfini, Alessio Lorusso, Giovanni Di Teodoro

**Affiliations:** 1Istituto Zooprofilattico Sperimentale dell’Abruzzo e Molise, 64100 Teramo, Italy; 2Faculty of Veterinary Medicine, University of Teramo, 64100 Teramo, Italy; 3Istituto Zooprofilattico Sperimentale della Puglia e della Basilicata, 71121 Foggia, Italy; 4Department of Veterinary Medicine, University of Bari, 70010 Bari, Italy

**Keywords:** D614G, respiratory ex vivo organ culture, minks, SARS-CoV-2, furin cleavage site deletion, viral replication, neutralization

## Abstract

SARS-CoV-2 has been shown to lose the furin polybasic cleavage site (FCS) following adaptation on cell culture. Deletion occurring in this region, which may include also the FCS flanking regions, seem not to affect virus replication in vitro; however, a chimeric SARS-CoV-2 virus without the sole FCS motif has been associated with lower virulence in mice and lower neutralization values. Moreover, SARS-CoV-2 virus lacking the FCS was shed to lower titers from experimentally infected ferrets and was not transmitted to cohoused sentinel animals, unlike wild-type virus. In this study, we investigated the replication kinetics and cellular tropism of a SARS-CoV-2 isolate carrying a 10-amino acid deletion in the spike protein spanning the FCS in lung ex vivo organ cultures of mink. Furthermore, we tested the neutralization capabilities of human convalescent SARS-CoV-2 positive serum samples against this virus. We showed that this deletion did not significantly hamper neither ex vivo replication nor neutralization activity by convalescent serum samples. This study highlights the importance of the preliminary phenotypic characterization of emerging viruses in ex vivo models and demonstrates that mink lung tissues are permissive to the replication of a mutant form of SARS-CoV-2 showing a deletion spanning the FCS. Notably, we also highlight the need for sequencing viral stocks before any infection study as large deletions may occur leading to the misinterpretation of results.

## 1. Introduction

The novel highly pathogenic coronavirus severe acute respiratory syndrome coronavirus 2 (SARS-CoV-2) (Coronaviridae Study Group of the International Committee on Taxonomy of Viruses, 2020) is responsible for the current worldwide pandemic known as coronavirus disease 2019 (COVID-19). Although SARS-CoV-2 is believed to be ancestrally linked to bats, the virus origin and possible intermediate host(s) have not yet been fully identified [1,2]. Several animal species have been demonstrated to be susceptible to SARS-CoV-2 either naturally or experimentally [3,4,5,6,7,8]. SARS-CoV-2 infection in American mink (*Neovison vison*) caused global concern resulting in the culling of millions of animals in Denmark [9]. This was largely due to the emergence of mink-derived SARS-CoV-2 strains containing multiple substitutions in the spike (S) glycoprotein encoding gene receptor binding domain (RBD) following multiple anthropozoonotic transmission events between humans and mink [10]. Some of these putative mink-related substitutions were subsequently identified in SARS-CoV-2 strains infecting humans with epidemiological links to the mink farms and thought to have impact on human angiotensin converting enzyme 2 (ACE2) binding affinity, as well as on humoral immune responses directed to the RBD region of the SARS-CoV-2 S protein [10].

In general, coronaviruses use their S glycoprotein to bind to its receptor and to induce host membrane cell fusion and virus entry [11]. The S protein, which contains two subunits (S1/S2), has to be primed by cleavage up to two times to perform its fusogenic activity with the host cell membrane [11]. The first priming cleavage between S1 and S2 site occurs during protein trafficking in the cells by host furin-like enzymes, or by serine-proteases at the cell surface following attachment, or by cathepsin proteases in the late endosome/lysosome, depending on the sequence at the S1/S2 junction. After the first cleavage and binding to the ACE2, a second cleavage site (CS) becomes exposed within the S2 domain (S2’ site). After the second cleavage, the S2 fusion peptide is liberated and triggers viral-cell membrane fusion [12]. A notable feature of SARS-CoV-2 is a polybasic cleavage site (RRAR) at the junction between S1 and S2, which is flanked by a leading proline (PRRAR) [13]. This allows effective cleavage by furin and other proteases and has a role in determining viral infectivity and host range, resulting in the addition of O-linked glycans to S673, T678 and S686, which flank the cleavage site and are unique to SARS-CoV-2.

Interestingly, SARS-CoV-2 has been shown in multiple independent studies to rapidly lose this polybasic CS upon multiple passages on monkey kidney epithelial E6 (Vero E6) cells, which represent the most popular cell line for virus isolation and propagation [14,15,16,17,18,19]. In this regard, SARS-CoV-2 variants with mutations at the S1/S2 CS are generated in vitro during propagation in transmembrane serine protease 2 (TMPRSS2)-deficient cells [20] including Vero E6 cells. Indeed, TMPRSS2 is essential for S protein priming triggering infection [21]. In addition, there are reports of CS mutants isolated directly from clinical swabs [16,18]. Several different mutants in this region are described, including total deletions of the CS, loss of arginine mutations within the CS making it less polybasic, or deletions of flanking regions leaving the polybasic tract intact but potentially affecting accessibility to protease [18]. Apparently, these deletions do not affect virus replication in vitro. However, it was demonstrated that a chimeric SARS-CoV-2 virus without the PRRA motif is less virulent in experimentally infected hamster and transgenic K18-hACE2 mice with respect to the parental strain [22]. The absence of the furin CS seems to be also correlated with lower neutralization values of both COVID-19 positive patient sera and monoclonal antibody directed against the RBD [22]. The presence of an intact furin cleavage site has been demonstrated to be critical also for transmission in ferrets [23]. In this study, we observed a deletion in the spike protein of a SARS-CoV-2 isolate spanning the furin cleavage site (Δfcs). In this concern, we tested the replication kinetics of this virus in ex vivo organ cultures (EVOCs) from mink lungs. Furthermore, we tested the neutralization capabilities of human convalescent serum samples against this mutant virus.

## 2. Results

By specific molecular and serological assays on rectal and nasopharyngeal swabs, neither nucleic acids of SARS-CoV-2, canine distemper virus (CDV) and aleutian mink disease virus (AMDV), nor antibodies against SARS-CoV-2 and CDV were detected. Viability of mink tissues was demonstrated by the marked replication of CDV (Figure 1 b,c). Both SARS-CoV-2 isolates, during the replication in mink lung EVOCs, displayed increase of viral titers on cell cultures (Figure 1b) and of the number of RNA copies (Figure 1c) over time. Titer values and number of RNA copies were significantly higher at 24, 48 and 72 h post infection (hpi), respectively, when compared to the earliest time point (*p* < 0.001). G614 replicated in mink lung EVOCs at higher magnitude at 48 hpi with respect to D614 Δfcs in both titrations (*p* < 0.05, Figure 1b,c).

IHC demonstrated SARS-CoV-2 antigens of both isolates in the cytoplasm of bronchial ciliated epithelial cells (Figure 1d,e). Immunoreactivity was not observed in mock-infected EVOCs. No differences of antigens distribution were observed between the two isolates. Mutations were not evidenced in the genome of viruses, sequenced out of explant supernatants with respect to viruses used for infection.

Apparently, in our experimental settings, the deletion in the spike protein spanning the furin cleavage site did not significantly hamper neutralization by convalescent serum samples (Figure 1f). Mean neutralization titers of 75.56 ± 37.62 and 102.8 ± 39.92 were calculated for D614 Δfcs and G614, respectively. In detail, 6/18 serum samples were shown to be negative for SARS-CoV-2 antibodies when tested against both viral isolates. In 6/12 positive serum samples, a 2-fold reduction in neutralization values was observed against D614 Δfcs with respect to G614; 1/12 showed instead a 4-fold reduction. The remaining five serum samples showed the same neutralization titers. However, differences between the two tested groups were not statistically significant (Figure 1f).

## 3. Discussion

In a previous study, we demonstrated that early SARS-CoV-2 variants were able to replicate in respiratory explants of cattle and sheep and that the isolate showing G614 in the spike protein exhibited enhanced replicative capabilities compared to the earlier D614 strain, originally collected from a Chinese tourist who visited Rome, Italy, in January 2020. However, after four passages on Vero E6 cells of the D614 virus, a 10-amino acid deletion, encompassing the furin cleavage site of the spike protein, was demonstrated by whole genome sequencing. This deletion was not observed on G614. The deletion was noticed as we planned to reproduce, by infecting mink lung explants, the experimental approach used previously on ruminant tissues [24]. Despite D614 Δfcs has a deletion encompassing the fcs along with a signature D614 both less advantageous features for replication, the growth kinetic in mink lung EVOCs was efficient and only a modest reduction was observed at 48 h with respect to G614 (*p* < 0.05). However, this finding was already evidenced between D614 and G614 in cattle lung tissues suggesting that this slight difference obtained in this round of experiments could be related to the presence of D614G rather than Δfcs. D614 Δfcs and G614 also exhibited the same immune reactivity in lung tissues of minks.

Neutralization was not significantly hampered by the deletion under investigation. However, although not significant, a slight reduction of neutralization titers was evident as previously observed also in other studies [22]. This evidence requires further studies by testing serum samples of vaccinees as an altered antibody neutralization profiles indicate a critical need to survey this mutation in the analysis of SARS-CoV-2 treatments and vaccines. Reasonably, further validations of our SN endpoint readout times are warranted to ensure that sensitivity of the assay is not affected by the replicative fitness of the deleted virus.

This study has certainly few weaknesses. First, the tested virus not only lacks the fcs but also its flanking regions. Therefore, in this perspective, further experiments are reasonably warranted in order to elucidate the role of fcs in the replication capabilities of SARS-CoV-2 in mink lung tissues. Second, it was not possible to test the role of this large deletion in the context of a D614 backbone due to the lack of a proper D614 viral stock. Third, as already stated in previous papers, experiments involving tissues explants are only suggestive of replication and tropism capabilities that need to be further investigated in vivo.

However, this study highlights the importance of the preliminary phenotypic characterization in ex vivo tissues and shows that mink lung tissues were permissive to the replication of a mutant form of SARS-CoV-2 showing a deletion spanning the furin cleavage site at the S1/S2 junction. Furthermore, we also showed that this deletion did not alter the neutralization capabilities of human SARS-CoV-2 convalescent serum samples. However, our data suggest also that this deletion is related with a slight reduction of neutralization values. Importantly, this report also highlights the importance of sequencing viral stocks before in vivo, in vitro and ex vivo studies as deletions may occur thus altering results and their interpretation. In this perspective, the use of human lung adenocarcinoma cells expressing TMPRSS2 (Calu-3) is strongly recommended to avoid the onset of undesired mutations in the spike protein encoding gene including the CS [25,26,27].

## 4. Materials and Methods

### 4.1. Ethical Approval

Human serum samples analyzed in this study were derived from the official monitoring activities for SARS-CoV-2 antibodies performed by the Local Public Health Authority of Abruzzo Region (Prot: 2020/0007891/GEN/GEN). Written consent was obtained from patients according to the Artt. 7 e 13 of Regolamento EU 2016/679.

### 4.2. Human Serum Samples

We evaluated a total of 18 sera that were available at the onset of the study and that had been collected from individuals who had been tested positive for SARS-CoV-2 RNA by qRT-PCR on human nasopharyngeal swabs, as previously described [28,29]. The set of sera was collected and stored at −20 °C until testing, then they were tested by serum neutralization assay (SN) to detect the presence of neutralizing SARS-CoV-2 antibodies. Unfortunately, we do not know with certainty at which time point, with respect the onset of symptoms, these serum samples were collected. Moreover, serum samples were collected from individuals either infected with alpha variant lineage or from other minor lineages clearly distinct from the prototype early Chinese strain.

### 4.3. Cell Culture

The Grivet monkey (*Cercopithecus aethiops*) kidney epithelial cell line Vero E6 (C1008) was kindly provided by Istituto Nazionale Malattie Infettive (INMI) Lazzaro Spallanzani, Rome, Italy. The cells were maintained in minimal essential medium (MEM, Sigma Aldrich, Merk Life Science S.r.l., Milan, Italy) supplemented with 10% fetal bovine serum (FBS, Sigma Aldrich, Merk Life Science S.r.l., Milan, Italy), 10^6^ IU/L penicillin, 10 g/L streptomycin, 5 × 10^6^ IU/L nystatin and 125 mg/L gentamicin (IZSAM). The cell line was regularly checked for *Mycoplasma* spp. contamination, and the absence was verified by PCR (Mycoplasma Detection Testing, Thermo Fisher, Waltham, MA, USA).

### 4.4. Animals

Lung tissue samples were collected from three 1 year-old American minks (*Neovison vison*, n = 3) belonging to a fur farm in Castel di Sangro, Abruzzo Region, Italy. Tissue sampling was performed in the accordance with the internal guidelines of the fur farm from animals which were regularly slaughtered. Additionally, respiratory and rectal swabs were collected before slaughtering and were tested for SARS-CoV-2 RNA (TaqMan^TM^ 2019-nCoV Assay Kit v2, Thermo Fisher, Waltham, MA, USA ), canine distemper virus (CDV) [30] (SuperScript III Platinum One-Step Quantitative RT-PCR System, Invitrogen, Waltham, MA, USA) and Aleutian mink disease virus DNA (AMDV) (Real-time PCR detection kit for Aleutian disease virus, Genesig, Primerdesign Ltd, York House, UK). Before tissue collection, serum samples had been collected from each animal and stored at −20 °C until testing. All sera were tested for the presence of neutralizing antibodies against SARS-CoV-2 (IZSAM) and CDV [25] by SN assay.

### 4.5. Virus Strains and Sequencing

Two SARS-CoV-2 isolates were selected for the experiments: (i) SARS-CoV-2/INMI1-Isolate/2020/Italy showing an in-frame 10-amino acid deletion (Δ_680_SPRAARSVAS_689_; Δfcs) in the spike protein encompassing the furin cleavage between S1 and S2 (Figure 1a); and (ii) SARS-CoV-2/IZSAM/46419 (GISAID accession number EPI_ISL_529023; hCoV-19/Italy/ABR-IZSGC-TE46419/2020, Pango lineage (https://pangolin.cog-uk.io/ (accessed on 24 August 2021)) B.1- Pango v.3.1.11). The non-deleted form of SARS-CoV-2/INMI1-Isolate/2020/Italy was kindly donated via EVAg (https://www.european-virus-archive.com/ (accessed on 24 August 2021)) by the INMI and its nucleotide sequence is available on the GISAID website (accession number EPI_ISL_410545; betaCoV/Italy/INMI1-isl/2020; Pango lineage B-Pango v.3.1.11). Both SARS-CoV-2 strains had been isolated and propagated on Vero E6 cells using MEM supplemented with 10% FBS. For great clarity, cells were seeded in 175 cm^2^ flasks at 10^6^ cells/mL and after 24 h were infected with 5 mL of a viral suspension at 0.01 multiplicity of infection. The flasks were incubated at 37 °C in a humidified atmosphere of 5% CO_2_ and observed daily under an inverted optical microscope. When cytopathic effect (CPE) affected 80–90% of the cell monolayer, the supernatant was collected and centrifuged at 4 °C 2000 rpm for 10 min to remove the cellular pellet. Then, the supernatant was aliquoted and stored at −80 °C. Before use, the virus was titrated in serial 1 log dilutions (from 1 log to 8 log) in 96-well culture plates of Vero E6 cells to determine the 50% tissue culture infective dose (TCID50). Plates were incubated at 37 °C and checked every day to identify CPE using an inverted optical microscope. The endpoint titers were calculated according to the Reed and Muench method based on 10 replicates for titration [31]. The high-throughput sequencing on both isolates showed that, with respect to the reference sequence of SARS-CoV-2 isolate Wuhan-Hu-1, the original SARS-CoV-2/INMI1-Isolate/2020/Italy provided by INMI shows G251V in the NS3, and SARS-CoV-2/IZSAM/46419 shows D614G in the spike protein. SARS-CoV-2/INMI1-Isolate/2020/Italy is, indeed, representative of the early Chinese isolates, whereas SARS-CoV-2/IZSAM/46419 is representative of the early Italian strains characterizing the so-called “first wave” of the Italian pandemic in late winter 2020 [32]. The deletion Δ_680_SPRAARSVAS_689_ spontaneously emerged starting from passage 4 on Vero E6 as all viral passages were monitored either by partial or whole genome sequencing before stocking. Supernatant showing (in mixed conditions) this deletion was plaque purified (pf) in order to obtain a pure viral stock (passage 1pf). Scrutiny of the deleted spike protein of D614 Δfcs for O-linked glycans (http://www.cbs.dtu.dk/services/NetOGlyc/ (accessed on 24 August 2021)) revealed the destruction of O-linked glycans to S673, T678 and S686 (Figure 1a) with respect to SARS-CoV-2/IZSAM/46419 and to prototype Wuhan-Hu-1 (NCBI Reference Sequence: NC_045512.2). SARS-CoV-2/IZSAM/46419 passage 3 was used for the infection experiments. Experiments with SARS-CoV-2 were performed in BSL-3 conditions as described previously [24,29]. Based upon the characterizing genome constellations and to facilitate reading, the deleted form of SARS-CoV-2/INMI1-Isolate/2020/Italy is indicated throughout the manuscript as D614 Δfurin cleavage site (D614 Δfcs), and SARS-CoV-2/IZSAM/46419 as G614. We were not able to obtain a sufficient amount of parental SARS-CoV-2/INMI1-Isolate/2020/Italy from previous viral stocks for in vitro studies, thus we decided to infect mink tissues with D614 Δfcs and compare its replication capabilities with G614.

### 4.6. Lung Ex Vivo Organ Culture (EVOCs)

Lung EVOC culture and viability were assessed as described previously [24,33,34]. Briefly, the right bronchus was catheterized and the lung parenchyma embedded with 1% low gelling temperature agarose (type VII-A agarose, Sigma-Aldrich, Merk Life Science S.r.l., Milan, Italy) at 37 °C. Then, the lung lobe was cooled to 4 °C for 20 min and cut into 1 mm thick slices. Tissue samples of about 25 mm^2^ were transferred into 6-well plates and cultured in an air–liquid interface system, slightly submerged in tissue culture medium. Lung EVOCs were infected with 10^3^ TCID_50_/mL of both SARS-CoV-2 isolates, by submerging the explanted tissues in 1 mL of each virus dilution. After 1 h of incubation, EVOCs were washed 3 times with phosphate-buffered saline (PBS) and then incubated in fresh tissue culture medium at 37 °C in 5% CO_2_ for 72 h. To study viral replication, 300 μL of supernatant was collected at 1, 24, 48 and 72 h post infection (hpi). Supernatants were subjected to measurement of number of viral genome copies over time and titration as previously described [22]. As for the control of tissue viability, lung tissues were also infected with a CDV isolate, CDV2784/2013 [35,36]. The two isolate stocks used for infection, RNAs purified from the supernatants collected at 24, 48, 72 and 96 hpi from infected EVOCs underwent whole genome sequencing, as previously described [37,38]. Analysis was performed by means of the SeqMan NGen of the Lasergene Genomics tool suite (DNASTAR, Inc., Madison, WI, USA).

### 4.7. Viral Immune Detection

Cellular tropism of both SARS-CoV-2 isolates was investigated by immunohistochemistry (IHC). To this purpose, three replicates of virus and mock-infected respiratory EVOCs were fixed in 10% neutral buffered formalin at 24 hpi. Three μm thick sections of EVOCs were dried at 37 °C, dewaxed and rehydrated in Bioclear (Bio Optica) followed by graded alcohol series. Antigen retrieval was performed by autoclaving at 121 °C for 15 min in EnVision™ FLEX Target Retrieval Solution Low pH (Dako Omnis). Sections were incubated at 4 °C overnight with a primary rabbit anti-SARS-nucleoprotein polyclonal antibody (Sino Biological; 40143-T62). Immunoreactions were detected using a peroxidase-conjugated polymer method and visualized using 3-3′-diaminobenzidine as chromogen (Dako REAL™ Envision). Negative controls were obtained with the omission of the primary antibody.

### 4.8. Serum Neutralization Assay

We used the SN assay to determine the neutralization activity of convalescent serum samples. For the assay we used one positive and one negative control serum, kindly provided by the Istituto Nazionale Malattie Infettive “Lazzaro Spallanzani” (INMI, Rome, Italy). Before testing, serum samples were inactivated by heating at 56 °C for 30 min. Two-fold serial dilutions (from 1:10 to 1:1280) of the tested 18 sera and the positive and negative control sera were prepared in 96-well plates using MEM supplemented with 2% FBS. Subsequently, an equal volume of 100 TCID_50_/mL of G614 and D614 Δfcs isolates was individually added to the diluted serum samples, and plates were incubated for 30 min at 37 °C in 5% CO_2_. After incubation, the serum-virus solutions were transferred to 96 well plates containing confluent Vero E6 cells seeded the day before. These plates were incubated for 72 h at 37 °C in 5% CO_2_ and were observed using an inverted microscope for a virus-specific cytopathic effect (CPE). The neutralization titer was defined as the reciprocal of the highest dilution without any viral CPE in the wells. Strain SARS-CoV-2/IZSAM/46419 (G614) is currently used as reference strain for quantitation of SARS-CoV-2 antibodies by SN at IZS-Teramo [39].

### 4.9. Statistical Analysis

As for the ex vivo experiments, sample sizes were calculated using the G*Power software (latest ver. 3.1.9.7, Heinrich-Heine Universitat, Düsseldorf, Germany). Results were expressed as means and standard deviation derived from three replicates of three independent experiments (one animal for each experiment). Multiple Student *t*-test was performed to compare sets of data. Differences were considered statistically significant when *p* value was ≤0.05. Comparison of neutralizing antibody titer by SN in 18 serum samples was performed with the non-parametric Mann–Whitney test. *p* values ≤ 0.05 were considered statistically significant. Analyses and graphs were obtained with GraphPad Prism 9 software (La Jolla, CA 92037, USA). Mean serum titers are expressed throughout the manuscript as the reciprocal of the highest serum dilution able to inhibit the virus CPE and standard error.

## Figures and Tables

**Figure 1 pathogens-11-01152-f001:**
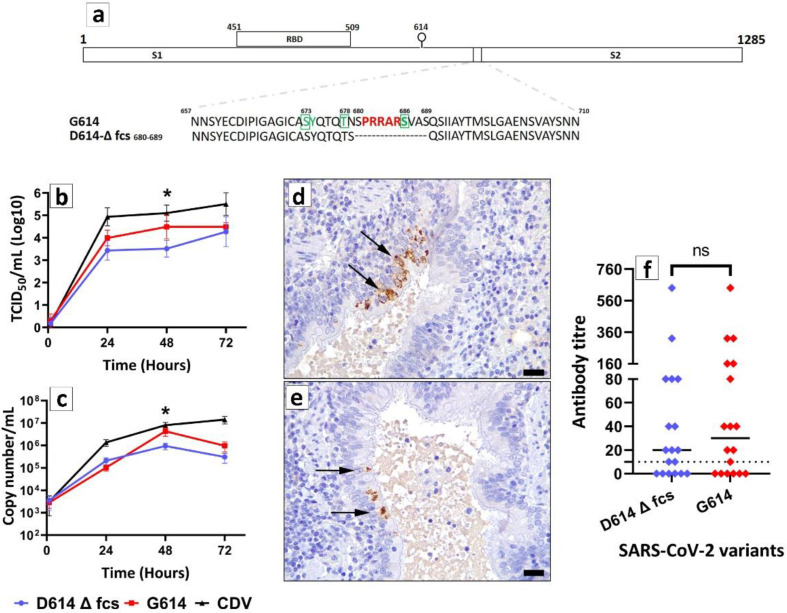
(**a**). Schematic representation of SARS-CoV-2 spike protein. RBD, receptor binding domain. S1 and S2, subunit 1 and 2, respectively, of the spike protein. G614, SARS-CoV-2/IZSAM/46419, that shows the complete furin cleavage site (highlighted in red) and its flanking regions. D614 Δfcs, SARS-CoV-2/INMI1-Isolate/2020/Italy that shows a 10-amino acid deletion (from position 680 to position 689). Green squares indicated O-linked glycans to S673, T678 and S686 present in G614 and destroyed in D614 Δfcs. A lollipop indicates residue 614. (**b**) SARS-CoV-2 replication kinetics in mink lung EVOCs. Viral titers expressed as Log10 TCID_50_/mL increased significantly, suggesting viral growth, from 1 to 72 hpi (*p* < 0.001). G614 replicated significantly at higher magnitude compared to D614 Δfcs at 48 hpi (*p* < 0.05). All EVOCs were infected with a dose of 10^3^ TCID_50_/mL of each SARS-CoV-2 isolate. Canine distemper virus (CDV) was used as positive control, with the same infectious dose. Titer values represent the mean of three replicates of three independent experiments. Bars indicate standard deviation. * (*p* < 0.05). Figure was obtained by GraphPad Prism 9 software (La Jolla, CA 92037, USA). (**c**) Quantification of SARS-CoV-2 RNA in mink lung EVOCs. Viral RNA levels expressed as copy number/mL increased significantly from 1 to 72 hpi (*p* < 0.001). G614 replicated in mink lung EVOCs at higher magnitude at 48 hpi with respect to D614 Δfcs in both titrations (*p* < 0.05). Values represent the mean of three replicates of three independent experiments. Bars indicate standard deviation. * (*p* < 0.05). Figure was obtained by GraphPad Prism 9 software (La Jolla, CA 92037, USA) (**d**,**e**) Immunohistochemistry for SARS-CoV-2 in mink lung EVOCs. SARS-CoV-2 nucleocapsid protein (arrows, brown color) was identified in the bronchial epithelium of infected mink lung EVOCs. Sections were counterstained with Mayer’s Hematoxylin (blue). Lung EVOCs infected with D614 Δfcs (**d**) and G614 (**e**) and formalin fixed after 24 hpi. Pictures were kept at final magnification of X400. Scale bar = 20 µm. (**f**) SARS-CoV-2 neutralizing antibody titers against G614 and D614 Δfcs isolates measured by serum neutralization. The mean values and standard error bars are presented. Antibody titer is plotted as the reciprocal of the highest serum dilution able to inhibit the virus cytopathic effect. Each dot represents one tested serum sample. The dashed line represents the cut-off value for positive test results. Comparison of neutralizing antibody titer by SN in 18 serum samples against G614 and D614 Δfcs was performed with the non-parametric Mann–Whitney test; ns = non-significant. Figure was obtained by GraphPad Prism 9 software (La Jolla, CA 92037, USA).

## Data Availability

The data presented in this study are available upon request to the corresponding author. Nucleotide sequences of viral strains are available on GISAID.

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
