# Peer review of "A Deletion Encompassing the Furin Cleavage Site in the Spike Encoding Gene Does Not Alter SARS-CoV-2 Replication in Lung Tissues of Mink and Neutralization by Convalescent Human Serum Samples"

_pathogens, 2022, doi:10.3390/pathogens11101152_

Round 1

Reviewer 1 Report

This is an interesting paper by Valleriani and colleagues that compared the replication kinetics between two SARS-CoV-2 isolates in ex vivo mink lung. One isolate has the 614G and the second possess the 614D and a 10 amino acids deletions spanning the furin cleavage site (Δfcs) acquired during cell culture. Authors also demonstrated that the neutralization activity of convalescent serum against the Δfcs strain was not significantly reduced. This short report aiming to analyse the importance of the furin deletion in immune escape and replication and tropism using also ex vivo models fulfil the '3Rs principles. 

In general, the paper is well designed, well written and easy to read. I recommend some minor comments as follows:

I recommend to check all the references of the introduction section. I don’t see ref n. 15 in the text and the citation of ref n. 19 is erroneous.

In figure legend and materials and methods I read the first time point is at 1 hpi so please change the “0” in the x axis of figure 1b-c with 1 hpi.

In figure 1e IHC signals seem to be non-specific. So you have to perform a control staining with omission of primary antibody and incubation with anti-rabbit IgG. In case of background staining I suggest to remove the e panel. 

Author Response

This is an interesting paper by Valleriani and colleagues that compared the replication kinetics between two SARS-CoV-2 isolates in ex vivo mink lung. One isolate has the 614G and the second possess the 614D and a 10 amino acids deletions spanning the furin cleavage site (Δfcs) acquired during cell culture. Authors also demonstrated that the neutralization activity of convalescent serum against the Δfcs strain was not significantly reduced. This short report aiming to analyse the importance of the furin deletion in immune escape and replication and tropism using also ex vivo models fulfil the '3Rs principles. 

In general, the paper is well designed, well written and easy to read. I recommend some minor comments as follows:

Q: I recommend to check all the references of the introduction section. I don’t see ref n. 15 in the text and the citation of ref n. 19 is erroneous.

R: Revised accordingly.

Q: In figure legend and materials and methods I read the first time point is at 1 hpi so please change the “0” in the x axis of figure 1b-c with 1 hpi.

R: Revised accordingly

Q: In figure 1e IHC signals seem to be non-specific. So you have to perform a control staining with omission of primary antibody and incubation with anti-rabbit IgG. In case of background staining I suggest to remove the e panel. 

R: Figure 1e has been replaced.

Reviewer 2 Report

1. Quite the opposite, this deletion was not ob-served on G614, although it was the reference strain used for SN assay, therefore with several serial passages on Vero E6. : This statement is unclear. It needs to be rephrased.

2. Despite D614 Δfcs has both a deletion and the less advantageous signature D614, the replication was efficient and only a modest reduction was observed at 48h.....: This statement as well.

Author Response

Q: Quite the opposite, this deletion was not ob-served on G614, although it was the reference strain used for SN assay, therefore with several serial passages on Vero E6. : This statement is unclear. It needs to be rephrased.

R: The statement has been rephrased.

Q: Despite D614 Δfcs has both a deletion and the less advantageous signature D614, the replication was efficient and only a modest reduction was observed at 48h.....: This statement as well.

R: The statement has been rephrased.

Reviewer 3 Report

The manuscript by Velleriani etal., has performed an interesting study on the role of furin cleavage site of SARS-Cov-2 spike protein on the viral replication and neutralization of convalescent human serum samples. The manuscript is well written, and experiments are performed well. Here are my comments for the authors which may help to improve the manuscript.

1. The authors should explain why the FCS deletion was observed specifically in the D614 strain but not in the G614 strain.

2. Authors should test the effect of FCS deletion in the D614 strain on the spike protein expression levels and cell surface localization in the cell culture experiments?

3. It would be more informative if authors align the amino acid sequences of some strains from the database which has the FCS deletion such as in the D614 strain. This may give some insights if there is any signature in the spike protein that may be leading to the FCS deletion.

Author Response

The manuscript by Velleriani etal., has performed an interesting study on the role of furin cleavage site of SARS-Cov-2 spike protein on the viral replication and neutralization of convalescent human serum samples. The manuscript is well written, and experiments are performed well. Here are my comments for the authors which may help to improve the manuscript.

Q: The authors should explain why the FCS deletion was observed specifically in the D614 strain but not in the G614 strain.

R: The deletion was observed by chance in D614 strain as this strain was passaged more times with respect to G614. Before infection of EVOC we wanted to make sure that the viral genome sequence was preserved as the low-passaged (and sequenced) strain. This deletion is rarely observed in vivo, it is rather a cell-culture dependent artifact. Overall, it is clearly stated in MM as this finding was evidenced.

Q: Authors should test the effect of FCS deletion in the D614 strain on the spike protein expression levels and cell surface localization in the cell culture experiments?

R: This is certainly a good point which will be exploited soon in our lab

Q: It would be more informative if authors align the amino acid sequences of some strains from the database which has the FCS deletion such as in the D614 strain. This may give some insights if there is any signature in the spike protein that may be leading to the FCS deletion.

R: A major published last year by Thomas Peacock et al analyzed 100,000 SARS-CoV-2 sequences derived from patients and 24 human postmortem tissues showed low frequencies of naturally occurring mutants that harbor deletions at the polybasic site. Although we deeply understand what this reviewer meant, this deletion may occur randomly in cell adapted strains regardless the genetic heterogeneity characterizing SARS-CoV-2 variants.

Reviewer 4 Report

In this study, the authors observed a deletion in the spike protein of a SARS-CoV-2 isolate spanning the furin cleavage site. They tested the replication kinetics of this virus in ex-vivo organ cultures (EVOCs) from mink lungs, and the neutralization capabilities of human convalescent serum samples against this mutant virus. They demonstrated that this deletion did not significantly hamper neither ex vivo replication and cellular tropism nor the neutralizing activity by convalescent serum samples. Overall, the manuscript is written in a clear and concise manner.

Minor Concerns:

1. Need indicators and scale bars in Fig 1. d. and Fig 1. e.

2. Statistical analysis of Fig 1. f. should be presented.

Author Response

In this study, the authors observed a deletion in the spike protein of a SARS-CoV-2 isolate spanning the furin cleavage site. They tested the replication kinetics of this virus in ex-vivo organ cultures (EVOCs) from mink lungs, and the neutralization capabilities of human convalescent serum samples against this mutant virus. They demonstrated that this deletion did not significantly hamper neither ex vivo replication and cellular tropism nor the neutralizing activity by convalescent serum samples. Overall, the manuscript is written in a clear and concise manner.

Minor Concerns:

Q: Need indicators and scale bars in Fig 1. d. and Fig 1. e.

R: Scale bars have been added.

Q: Statistical analysis of Fig 1. f. should be presented.

R: Revised accordingly.